# Norovirus Infections and Disease in Lower-Middle- and Low-Income Countries, 1997–2018

**DOI:** 10.3390/v11040341

**Published:** 2019-04-10

**Authors:** Janet Mans

**Affiliations:** Department of Medical Virology, Faculty of Health Sciences, University of Pretoria; Pretoria 0001, South Africa; janet.mans@up.ac.za; Tel.: +27-12-319-2660

**Keywords:** norovirus, lower-middle-income countries, low-income countries, genotype diversity, GII.4, systematic review

## Abstract

Noroviruses are a major cause of viral gastroenteritis. The burden of the norovirus in low-resource settings is not well-established due to limited data. This study reviews the norovirus prevalence, epidemiology, and genotype diversity in lower-middle-income countries (LMIC) and in low-income countries (LIC). PubMed was searched up to 14 January 2019 for norovirus studies from all LIC and LMIC (World Bank Classification). Studies that tested gastroenteritis cases and/or asymptomatic controls for norovirus by reverse transcription-polymerase chain reaction (RT-PCR) were included. Sixty-four studies, the majority on children <5 years of age, were identified, and 14% (95% confidence interval; CI 14–15, 5158/36,288) of the gastroenteritis patients and 8% (95% CI 7–9, 423/5310) of healthy controls tested positive for norovirus. In LMIC, norovirus was detected in 15% (95% CI 15–16) of cases and 8% (95% CI 8–10) of healthy controls. In LIC, 11% (95% CI 10–12) of symptomatic cases and 9% (95% CI 8–10) of asymptomatic controls were norovirus positive. Norovirus genogroup II predominated overall. GII.4 was the predominant genotype in all settings, followed by GII.3 and GII.6. The most prevalent GI strain was GI.3. Norovirus causes a significant amount of gastroenteritis in low-resource countries, albeit with high levels of asymptomatic infection in LIC and a high prevalence of coinfections.

## 1. Introduction

Noroviruses are a major cause of sporadic and epidemic gastroenteritis worldwide [1]. Severe disease is most often observed in young children (<5 years of age) [2], the elderly population (>65 years of age) [3], and immunocompromised individuals [4]. Noroviruses form a genus of the *Caliciviridae* family and are divided into at least seven genogroups (GI–GVII) [5]. Of these GI, GII, and GIV infect humans, with GII detected most frequently in clinical surveillance studies throughout the world [6]. The genogroups are subdivided in >30 genotypes. The predominant strain causing both outbreaks and sporadic cases of gastroenteritis is GII.4 [7,8]. Novel variants of GII.4 emerge periodically, spread rapidly across the globe, and cause pandemics [9]. Recently, other genotypes such as GII.17 and GII.2 have emerged and have become predominant in certain regions of the world [10,11]. A comprehensive understanding of the global burden of norovirus illness as well the genotype diversity, dominant strains, and strain replacement patterns is essential for a successful vaccine design [12].

In 2008, Patel and colleagues estimated the global norovirus prevalence at 12% (95% CI 10–15) based on 31 studies published between January 1990 and February 2008 [13]. More recently, the norovirus prevalence was estimated at 18% (95% CI 17–19) based on 175 studies spanning January 2008 to March 2014 [1]. Norovirus-associated infections are estimated to annually cause 677 million cases of disease and approximately 200,000 deaths globally [14]. The majority of these deaths occur in low-resource countries. Several systematic reviews have reported the prevalence of the norovirus in specific regions such as Latin America (15%; 95% CI 13–18) [15,16], China (19.8%–21.0%) [17], the Middle East and North Africa (MENA) (15.3%—median infection rate) [18], and 46 low-resource (developing) countries (17%; 95% CI 15–18) [19].

On the whole, there is still a lack of data on the norovirus prevalence in low-income countries (LIC). The World Bank classifies countries into one of four categories (high income countries, upper-middle-income countries (UMIC), lower-middle-income countries (LMIC), and LIC) based on their annual gross national incomes. The majority of data used to estimate global norovirus prevalence emanated from studies performed in high- and upper-middle-income countries [1,13,19]. In recent years, there has been an increase in studies from low-resource settings [20,21,22,23,24,25,26,27,28], reflecting a growing awareness of the importance of noroviruses as enteropathogens. Studies on the burden of norovirus are complicated by the frequent occurrence of asymptomatic infections [29], a significant proportion of norovirus disease that manifests with vomiting only symptoms [30], and a high prevalence of coinfections with other gastroenteritis pathogens [31].

The Etiology, Risk Factors, and Interactions of Enteric Infections and Malnutrition and the Consequences for Child Health and Development (MAL-ED) cohort study evaluated the aetiology of diarrhoea in children 0–2 years old in community settings in eight low-resource locations [32]. Molecular diagnostic assays were used to screen 6625 diarrhoeal and 30,968 control stools from 1715 children for 29 enteropathogens [33]. Viral diarrhoea was found to be more common than bacterial diarrhoea, and ten pathogens were responsible for 95% of diarrhoeal diseases. Norovirus was among the top ten pathogens and was found to cause 15.4 attributable diarrhoea episodes per 100 child-years, 95% CI 13.5–20.1, with the highest incidence in children <11 months of age. Another study utilized the Global Rotavirus Surveillance Network to investigate the aetiology of severe acute watery diarrhoea [34]. Operario and coworkers screened 878 acute watery diarrhoeal samples collected from children <5 years of age from 16 countries (8 LIC, 5 LMIC, and 3 UMIC) for multiple pathogens by polymerase chain reaction (PCR). Norovirus GII was the third most commonly detected pathogen after rotavirus and enteroaggregative *E. coli* with an overall attributable fraction of 6.2 (95% CI 2.8–9.2). These studies show that noroviruses are important pathogens globally, but there is still a need for more data on the prevalence and diversity of noroviruses in LICs.

This review focused on studies published between 1997 and 2018 on norovirus prevalence, epidemiology, and diversity in LIC and LMIC and includes 64 studies of which 17 (5 LIC and 12 LMIC) were published after March 2016 and have not been included in previous systematic reviews. The aim of the study is to review the norovirus prevalence in symptomatic and asymptomatic cases; to identify the predominant genotype circulating in LIC and LMIC; and to summarise the clinical features, seasonality, and pathogen coinfections of norovirus infections in low-resource settings.

## 2. Methods

### 2.1. Search Strategy and Inclusion Criteria

The PubMed database was searched up to 14 January 2019 with the following search terms: “norovirus”, “calicivirus”, and “*Caliciviridae*” in combination with the names of each of the LIC (34) and LMIC (47) countries as defined by the June 2018 World Bank list of economies [35]. The search filter was set to exclude hits containing the term “outbreak” or “outbreaks” in the title (see Appendix A), since the review was aimed at sporadic norovirus disease. Only studies published in English were considered. Studies were included if they detected the norovirus with reverse transcription (RT)-PCR-based methods in individuals with acute gastroenteritis symptoms and/or asymptomatic control groups or populations. No limitations were set on the study size, duration, or age group of the study population. Studies which reported on norovirus detection in consecutive samples of cases and/or controls were excluded.

### 2.2. Data Extraction

The following information was extracted from each included study: First author; year of publication; study duration; study setting (i.e., inpatient or outpatient/community); age group; number of cases screened; number of cases positive for the norovirus; number of control subjects screened for the norovirus; number of control subjects positive for the norovirus; age information for norovirus positive cases and controls; number of norovirus GI, GII, and GIV detections if available; clinical symptoms associated with norovirus detection; coinfections with other pathogens; genotype information; and data on seasonality. Where GenBank accession numbers were provided, the sequences were analysed by the online norovirus genotyping tool (https://www.rivm.nl/mpf/typingtool/norovirus/) to confirm the genotypes. The data was stratified according to LIC and LMIC, hospitalized and outpatient/community groups, combined inpatient and outpatient data, age groups of 0–<5 years, 0–18 years, mixed ages, and asymptomatic infections including all age groups.

### 2.3. Statistical Analysis

The web-based open-source statistical calculator OpenEpi version 3 (https://www.openepi.com) was used to calculate the 95% confidence intervals for proportions [36]. Statistically significant differences between the norovirus proportions in different settings were compared with a *z*-test using EpiTools epidemiological calculators (http://epitools.ausvet.com.au/content.php?page=z-test-2).

## 3. Results

The PubMed literature search identified 352 articles, of which 331 titles and abstracts were screened for possible inclusion. Two hundred and twenty-five nonrelevant articles were excluded, and 106 full text articles were evaluated for inclusion. Forty-two of the 106 articles were excluded for reasons listed in Figure 1. Sixty-four studies were included in the review of the norovirus prevalence. Forty-three studies, in total, were included for a norovirus genotype analysis. Of these, 37 were studies that reported both the prevalence and genotype data, and 6 were studies that reported the genotyping data from noroviruses detected in pathogen-negative specimens (Figure 1).

Sixty-four studies representing eight LIC (15 studies) and 21 LMIC (49 studies) (see Appendix A) were included in the review of the norovirus prevalence (Figure 2). Additional studies from countries which tested rotavirus-negative specimens for noroviruses were included in the world map with their respective norovirus prevalence. The median size of the study populations in LIC was 250, and studies ranged from very small (48 participants) to large (2678 participants). Studies in LMIC had a median sample size of 265 and ranged from 34 to 2495 participants. The majority of LMIC studies were from India (13), Vietnam (7), and Bangladesh (5), whereas between one to three studies from each LIC were included.

### 3.1. Norovirus Prevalence in LIC and LMIC

Overall, across all age groups in LIC and LMIC, norovirus was detected in 14% (95% CI 14–15) of individuals with gastroenteritis symptoms and in 8% (95% CI 7–9) of asymptomatic controls. Norovirus was detected more frequently in symptomatic cases of gastroenteritis in LMIC (15%; 95% CI 15–16, range 0–43%) than in LIC (11%; 95% CI 11–12, range 6–42%) (Table 1). In asymptomatic controls, noroviruses were detected in 9% (CI 8–10, range 3–12%) in LIC and in 8% (CI 7–9, range 0–19%) in LMIC studies. In both the LIC and LIMC settings, there was a statistically significant difference in the norovirus prevalence between children (<5 years of age) that were hospitalized (LMIC: 17% CI 17–18; LIC: 11% CI 10–12) and nonhospitalized children (outpatients/community) (LMIC: 12% CI 10–14; LIC: 9% CI 8–11; LMIC *p* < 0.0001 and LIC *p* = 0.0058).

From 37 studies in children <5 years of age, norovirus was detected most frequently in children <1 year of age in five studies and in children <2 years of age in 16 studies. An additional 10 studies reported the median age of the norovirus-positive population, ranging between 6–15.5 months of age.

The prevalence of norovirus GI in relation to GII was similar in symptomatic infections in both settings (Table 2). The increased detection of norovirus GI was reported in asymptomatic controls in both LIC and LMIC (21.6% and 24% vs. 12% and 13% in symptomatic cases). Only three studies from Bangladesh screened stool specimens for norovirus GIV, and this genogroup represented 3% of the norovirus positives in these studies.

### 3.2. Norovirus Clinical Features

Most of the studies that presented data on clinical symptoms reported vomiting, fever, and abdominal pain in patients with diarrhoea. In 20 studies, the percentage of patients with vomiting ranged between 37.5% and 92%, with a median of 68%. There was a larger variation in the number of patients with fever. Within 16 studies, the percentage of patients with a fever ranged from 5% to 84%, with a median of 46%. Abdominal pain was reported in six studies, and the median percentage of patients with this symptom was 57% (ranging from 46% to 65%). Since data was not provided on the combination of symptoms in patients, it is not possible to estimate how often these symptoms occur as single symptoms or in combination. Six studies reported severe dehydration in the following percentages of single norovirus infections, 15.5% [26], 18.9% [77,97], 19% [48], 19.4% [38], and 30.3% [71]. Mild dehydration was observed in a larger number of the patients, 37.8% [97] and 60% [26]. Some studies categorised dehydration as moderate to severe and reported between 2.9% and 38% of patients with this level of dehydration. 

### 3.3. Norovirus and Coinfections

The studies included in this review that reported on coinfections of norovirus with other gastroenteritis pathogens can be divided into three types: (a) studies that tested for norovirus and rotavirus (n = 12); (b) studies that tested for a range of gastroenteritis viruses (n = 12); and (c) studies that tested for enteric viruses, bacterial pathogens, and/or parasites (n = 8). Figure 3 provides an overview of the norovirus coinfections described in 26 studies from 15 LMIC and 6 studies from 4 LIC. Overall, these studies detected the norovirus in 2899 cases, of these 2131 (73.5%) were single norovirus infections and 768 (26.5%) were coinfections with various gastroenteritis pathogens. The majority were norovirus/rotavirus coinfections (18.3%), and 8.2% represented coinfections with other pathogens. The percentage of coinfections detected in the different studies ranged from 5% to 76.7%, with a median of 26.7%.

### 3.4. Norovirus Seasonality

The LIC and LMIC in this review are generally located in the tropical or subtropical regions of the world. Thirty-eight of the LMIC studies and seven LIC studies reported data on seasonality. In the majority of studies, norovirus was detected throughout the year with peaks of norovirus activity varying between months or seasons. To determine seasonal trends, the studies were grouped into the following geographical regions: Asia, Southeast Asia, North Africa/Middle East, West Africa, East Africa, and Central and South America. However, no clear seasonality was apparent from the different studies when grouped into these regions. Two studies in Nicaragua both reported a peak in norovirus detection during the early rainy season in June and July [21,77]. The three countries that published multiple norovirus studies are Bangladesh, India, and Vietnam. One Bangladesh study [28] reported norovirus detection throughout the year, with a peak in the hot summer months of March to April. The second Bangladesh study spanned three years and reported a slightly higher detection rate in the winter and summer seasons compared to the rainy season [29]. Another Bangladesh study performed between October 2004 and September 2005 detected a peak of norovirus activity between January and March 2004, which corresponded to the cooler months of the year [37]. No clear trend is discernable from these studies. A two-year study in western India by Chhabra and coworkers [54] showed a peak of norovirus detection in March, which coincided with decreased humidity, minimal rainfall, and higher temperatures. The other studies from India did not provide clear data on seasonality. Three studies from southern Vietnam reported peak norovirus detection in the rainy season [89,98] or towards the end of the season [99]. According to Tra My and colleagues there was a positive linear correlation between rainfall and norovirus infections [98].

### 3.5. Norovirus Genotype Diversity

Norovirus genotypes are classified with a phylogeny-based dual-typing system using partial polymerase and complete capsid nucleotide sequences (i.e., GII.Pe/GII.4) [100]. Overall, 37/64 studies reported norovirus capsid genotyping data (Appendix A). An additional six studies with capsid genotyping data were included in the genotype diversity analysis. These were excluded from the prevalence review because only rotavirus-negative stool specimens were screened [87,88,99,101,102,103]. Ten out of 15 LIC studies from six countries included in the review reported norovirus genotyping data (Figure 4a). The studies were conducted between 1997 and 2013, and 30 genotypes (9 GI, 1 unassigned GI and 20 GII) were described, with GII.4 having the widest distribution (6 countries) followed by GI.3, GII.2, and GII.6 that were reported from 5 countries. Norovirus GI.5, GII.3, GII.10, and GII.14 were detected in four countries. The genotypes that were found in three countries were GI.4, GI.7, GII.7, GII.16, and GII.17.

Norovirus genotyping data were reported in 27/49 LMIC studies from 14 countries (Figure 4b). These studies were published between 2004 and 2018 and represent a range of timeframes between 1997–2015. Overall, 28 genotypes (9 GI, 19 GII) were circulating in the LMIC. As with LIC, GII.4 was detected in all of the countries (14), with GII.2 and GII.3 being the next most widely distributed genotypes reported in 11 countries; GII.6 was identified in 10 countries, GII.1 in 9 countries, and GII.7, GII.14 and GII.16 were identified in 8 countries.

Overall, the most common and widely distributed genogroup I strain was GI.3 followed by GI.7, GI.5, GI.2, and GI.1. Among the genogroup II strains, GII.4 was the most frequently detected in a total of 1490 cases (250 LIC and 1240 LMIC), with GII.3 as the second most dominant strain identified in 389 cases. The other strains that were detected in multiple countries in at least 50 cases are GII.6, GII.2, GII.21, GII.13, and GII.1.

Norovirus GII.4 variants are distinguished based on a phylogenetic analysis of the complete capsid gene [100]. New variants are recognised once they have been established in at least two separate geographic locations [100]. Fourteen different norovirus GII.4 variants were reported between 1997 and 2015 in LIC and LMIC (Figure 5). In LIC, Yerseke 2006a, Osaka 2007, New Orleans 2009, and Sydney 2012 represented the majority of variants. In LMIC, Hunter 2004, Den Haag 2006b, New Orleans 2009, and Sydney 2012 were the most frequently reported.

## 4. Discussion

The World Bank list of economies classified 81 countries as LMIC and LIC in June 2018. Of these, 29 were represented in this review of norovirus prevalence and diversity in low-resource countries. Overall, 36,288 symptomatic gastroenteritis cases were tested for norovirus in 64 studies over a period of 19 years. Twenty-three of the 64 studies tested 5310 healthy controls for norovirus. Across all ages, norovirus was detected in 14% (95% CI 14–15, 5158/36288) of symptomatic cases and in 8% (95% CI 7–9, 423/5310) of healthy control subjects. This correlates with previous global prevalence estimates of 14% (95% CI 11–16) in symptomatic cases in high-mortality developing countries and 7% (95% CI 3–10) in asymptomatic controls [1]. When countries were grouped according to income, norovirus was detected across all ages in 11% (95% CI 11–12) of symptomatic cases and 9% (95% CI 8–10) of asymptomatic controls in LIC. The detection rate for both symptomatic and asymptomatic norovirus infections in LIC was higher than the 6% (95% CI 3–10) and 5% (95% CI 4–8) estimated previously [19]. This is likely due to the additional data included in this review. The five studies from LIC that were published since 2016 had a mean norovirus prevalence of 15% and contributed 224/1455 norovirus positive cases to the current review. This may also reflect the improved detection via quantitative RT-PCR, as four of the five studies used quantitative versus qualitative RT-PCR [20,22,27,81], which was used in many earlier studies in LIC. In the current review, norovirus was detected in LMIC in 15% (95% CI 15–16) of symptomatic cases and 8% (95% CI 7–9) of healthy controls. The percentage of norovirus detected overall in symptomatic cases correlates well with previous estimates in LMIC (15%; 95% CI 13–18) [19], in Latin America (15%; 95% CI 13–18) [15], and in MENA countries (15.3% median) [18]. The asymptomatic detection rate of 8% is comparable to the previous LMIC (5%; 95% CI 3–10) [19] and Latin American (8%; 95% CI 4–13) estimates [15]. These data are in agreement with a seroepidemiological study in Uganda [104], which detected norovirus antibodies in 77% of children by 1 year of age and in 99% of two-year old children and strongly suggest that the rate of asymptomatic infections in infancy and early childhood is high in low-resource settings.

More than half of the included studies screened for noroviruses in children <5 years of age. Due to the relatively small number of outpatients and patients in community settings, these groups were combined and compared with hospitalized children. In LIC, 11% (95% CI 10–12) of inpatient children tested positive for norovirus, whereas 9% (95% CI 8–11) of outpatients tested positive (Table 1). In LMIC, a larger difference was observed with a 17% (95% CI 17–18) norovirus detection in inpatients <5 years of age and only 12% (95% CI 10–14) in outpatients of the same age. Both differences were statistically significant. In contrast, Ahmed and coworkers [1] found that the norovirus prevalence was higher in community (24%; 95% CI 18–30) and outpatient (20%; 95% CI 16–24) settings, compared to inpatients (17%; 95% CI 15–19). Their study included a smaller number of studies from low-resource countries and were classified according to mortality rates rather than income, which may explain the differences. In Latin America, the norovirus prevalence was comparable between inpatients (16%; 95% CI 12–21) and outpatients (14%; 95% CI 10–19) and in the community (15%; 95% CI 13–18) [15].

Norovirus GII was the predominant genogroup detected in all studies in LIC and LMIC. Similar GI and GII distributions were reported (GI/GII/mixed: 12/87/0.9, 13/83/3) in symptomatic infections. Only three studies in Bangladesh screened for GIV and detected it at approximately 3% of gastroenteritis cases. Norovirus GI occurred more frequently in the asymptomatic control population in both LIC (22:78 GI:GII) and LMIC (24:73 GI:GII). This correlates with norovirus-specific data from the MAL-ED study [105] which showed that GII infections had an 85% increased odds of being symptomatic compared to GI infections.

The inclusion criteria for 87% of the LMIC studies were the classic diarrhoea definition of ≥3 loose stools in the last 24 h. The clinical features of norovirus infections, in addition to diarrhoea, reported most often were vomiting and fever. Seven studies added “with or without vomiting” to their definition of gastroenteritis, but only one study stated vomiting only cases in addition to diarrhoeal cases in their inclusion criteria. Therefore, as with the majority of norovirus surveillance studies, vomiting only cases were missed, which suggests that the prevalence of the norovirus might be underestimated. The classification of dehydration varied between studies, and since few actually reported on dehydration, it is not possible to estimate the overall level of dehydration that norovirus infections caused in the LIC and LMIC settings. Six studies reported severe dehydration in 15–30% of single norovirus positive cases, and mild dehydration was observed in up to 60% of norovirus infections in a recent study in Cameroon [26]. Four studies provided comparative data on dehydration for norovirus and rotavirus. In two of these, rotavirus caused dehydration in a higher percentage of patients (64% vs. 18.9% [77] and 56% vs. 30% [71]) than norovirus. In a study in Ethiopia, norovirus caused dehydration in 19% of single norovirus infections compared to only 5% of rotavirus infections that led to severe dehydration [48]. Another study in Zambia reported comparable levels of mild and moderate dehydration for patients infected with rotavirus or norovirus but no severe dehydration [96]. Thus, the limited available data suggest that norovirus could display an increased severity in some settings, but in general, rotavirus causes a more severe disease.

With the advent of multiplex testing for large pathogen panels, widespread coinfections with enteropathogens have been observed in many studies [68,106,107,108]. Approximately a third of the studies (11/32) that reported coinfections in this review tested only for rotavirus and norovirus. Thus, the median of 26.7% coinfections is most likely an underestimation, since the percentage of coinfections will increase with the testing of more pathogens. Another third of the studies (12/32) tested for additional enteric viruses, and among these, six studies still only detected norovirus/rotavirus coinfections. One study from India [52] that tested for a large panel of pathogens also only detected norovirus and rotavirus infections and coinfections. In three studies, one on hospitalized neonates in India [60] and two others in India [55] and Pakistan [79], the rotavirus coinfections equaled or exceeded the single norovirus infections. The highest percentage of coinfections were detected in studies from Burkina Faso (65.5%, [27]), India (77%, [60]), and Tunisia (74.2%, [88]). Two recent studies on coinfections in diarrhoeal disease based on symptomatic and asymptomatic study populations from Rwanda and Zanzibar [109] and Tanzania [31] detected 65% and 58.1% of coinfections in symptomatic cases. Andersson and coworkers [109] found a negative association (coinfections less common than expected from probability) between rotavirus and norovirus GII in symptomatic patients. Statistically, pathogens that independently cause diarrhoea are expected to be negatively associated in symptomatic patients [109]. The other study reported four pathogens, rotavirus, norovirus GII, *Cryptosporidium*, and *Shigella* species/EIEC to be associated with diarrhoea in both mono-infections and coinfections [31]. Both these studies provide evidence for norovirus GII as a pathogen that causes diarrhoea in both mono- and coinfections.

In the northern hemisphere, norovirus has a very clear winter seasonality [110]; however, in other regions of the world, the seasonality is less well-defined. In this review, two countries, Nicaragua and Vietnam, reported a peak norovirus detection during the rainy season. Bucardo and colleagues provided data from Nicaragua between 1999 and 2015 that showed a clear peak of norovirus detection in June and July [21]. Furthermore, a one-year longitudinal study on norovirus in children <5 years of age in Nicaragua reported that norovirus infections peaked early in the rainy season (April–June) [111]. Data from Vietnam suggest that norovirus infections are positively correlated with rainfall, and three different studies detected norovirus peaks during the rainy season. In a systematic review on norovirus seasonality, Ahmed and coworkers found an association between norovirus season strength and average rainfall in the wettest month for outbreaks but not for sporadic cases [110]. Between 2002–2007 in Victoria, Australia, the norovirus outbreaks peaked approximately three months after the highest average rainfall [112]. The mechanism by which rainfall impacts on norovirus infections is not understood but may be linked with the environmental spread of the virus. Norovirus seasonal patterns are determined by complex factors, and it is clear that additional larger studies spanning several years are needed to understand seasonal distribution in tropical and subtropical low resource regions.

The norovirus genotype distribution in LIC and LMIC were very similar, with GII.4 overall detected at the highest frequency (1490/3233 typed strains) and with the widest distribution (20 countries). Norovirus GII.3 was clearly the second most prevalent genotype (389/3233 typed strains) and circulated in 15 countries. The third most prominent genotype was GII.6 (131/3233 typed strains), which was detected in 16 countries. Although GII.4 predominated in general, there were instances in Bhutan [41], Ethiopia [48], India [60], and Tunisia [86] where GII.3 was more prevalent than GII.4, and in 2001–2012, GII.21 was the most prevalent type detected in Bhutan [42]. Since the studies that form part of this review were performed between 1997 and 2015, the more recent emergence of novel GII.17 [11,113] and GII.2 [10] in LIC and LMIC could not be evaluated. Overall, the data correlate well with those reported by Hao-Tran and colleagues [8] in a review on the global distribution of norovirus genotypes in children with sporadic gastroenteritis, where GII.4 and GII.3 were the predominant genotypes circulating in children globally. The well-known predominance of GII.4 may be related to its ability to evade the immune system by the conformational occlusion of blockade antibody epitopes [114] and the variation in epitopes that allow an escape from immunity without impairing binding to cellular ligands [115]. There is a lack of genogroup I genotyping data globally, since GI strains are generally detected in less than 20% of sporadic norovirus cases. The 43 studies that reported genotypes from LIC and LMIC characterised 236 GI strains. The GI.3 strain predominated at 31% with GI.7 (15%), GI.2 (10%), and GI.1 (9%) among the other prevalent strains. 

The GII.4 variant distribution differed slightly between LIC and LMIC; whereas New Orleans 2009 and Sydney 2012 were predominant in both settings, in LIC, Yerseke 2006a and Osaka 2007 were also commonly reported but Hunter 2004 and Den Haag 2006b were more prevalent in LMIC. These patterns may reflect the varying time frames of the different studies rather than the predominance of a particular variant in a certain region.

There are several limitations to this review since the studies were quite heterogeneous. In general, the studies had small population sizes and some were of a short duration, which specifically impacted the conclusions on seasonality. The level of reporting varied greatly between different studies; thus, a full picture of the clinical presentation and coinfection levels could not be obtained.

## 5. Conclusions

An improved understanding of the norovirus epidemiology in low-resource settings is essential for successful vaccine development and implementation of interventions to counter the detrimental consequences of multiple gastroenteritis episodes early in life. The recent data on norovirus detections in LIC and LMIC correlate with the earlier estimates on the norovirus prevalence in these settings and emphasizes that noroviruses are important pathogens in low-resource countries. The predominance of norovirus GII.4 and GII.3, at least in the paediatric population is very clear, with a great diversity of genotypes circulating at a low frequency. However, to date, there are molecular-based norovirus data available for only 29 out of 81 low-resource countries, and more studies from various regions and in different age groups are necessary to aid strategies to combat norovirus gastroenteritis.

## Figures and Tables

**Figure 1 viruses-11-00341-f001:**
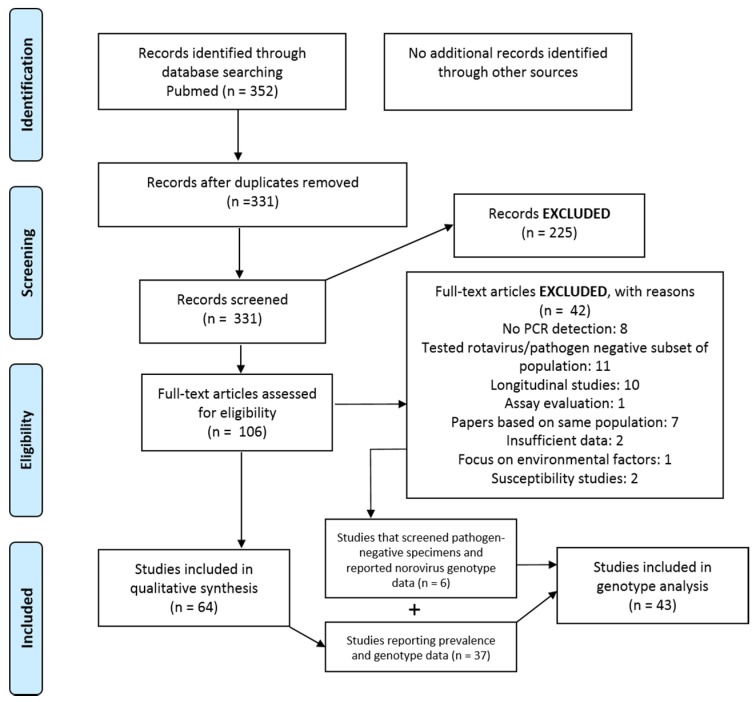
The study selection flow diagram.

**Figure 2 viruses-11-00341-f002:**
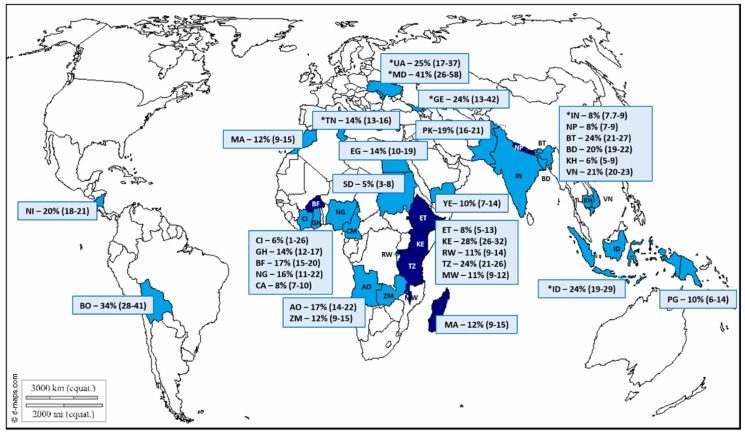
The world map indicating the low-income countries (LIC; dark blue) and lower-middle-income countries (LMIC; light blue) that are represented by the norovirus studies. The average norovirus prevalence and 95% confidence interval (95% CI) is indicated for each country. Countries are identified by two-letter International Organization for Standardization (ISO) codes, and the study references are indicated in brackets. AO, Angola [23]; BD, Bangladesh [28,29,37,38,39]; BO, Bolivia [40]; BT, Bhutan [41,42]; BF, Burkina Faso [20,27,43]; CI, Cote d’Ivoire [44]; CM, Cameroon [26]; KH, Cambodia [45]; EG, Egypt [46]; ET, Ethiopia [47,48]; GE, Georgia* [49]; GH, Ghana [50,51]; IN, India [52,53,54,55,56,57,58,59,60,61,62,63,64,65]; ID, Indonesia [66,67]; KE, Kenya [68]; MG, Madagascar [69]; MA, Morocco [70,71]; MW, Malawi [72,73]; MD, Republic of Moldova* [49]; NP, Nepal [22,74]; NG, Nigeria [75,76]; NI, Nicaragua [21,77]; PG, Papua New Guinea [78]; PK, Pakistan [79,80]; RW, Rwanda [81]; SD, Sudan [82]; TZ, Tanzania [83,84,85]; TN, Tunisia [86,87,88]; UA, Ukraine* [49]; VN, Vietnam [24,89,90,91,92,93,94]; YE, Yemen [95]; ZM, Zambia [96]. (https://d-maps.com/carte.php?num_car=13181&lang=en). *Includes data from studies that screened pathogen-negative stool specimens for norovirus.

**Figure 3 viruses-11-00341-f003:**
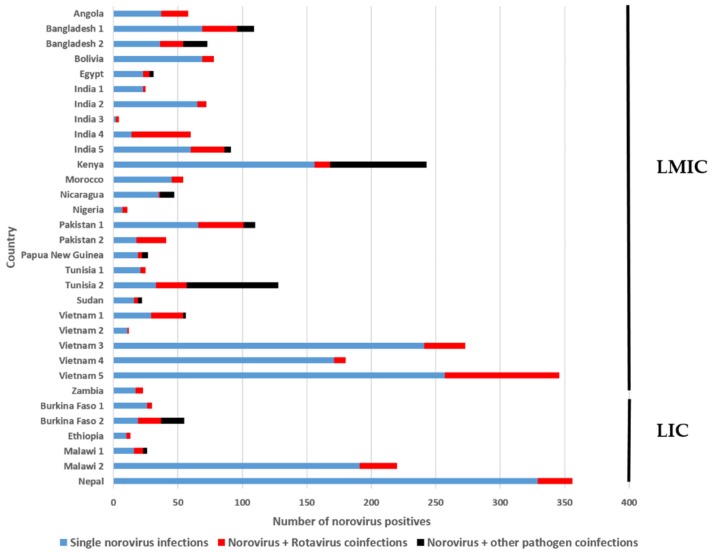
An overview of the norovirus coinfections from 26 studies in 15 LMIC countries and six studies in four LIC countries: Twelve studies tested for norovirus and rotavirus; 12 studies tested for a range of enteric viruses; five studies tested for viruses and bacteria; and three studies tested for viruses, bacteria, and parasites.

**Figure 4 viruses-11-00341-f004:**
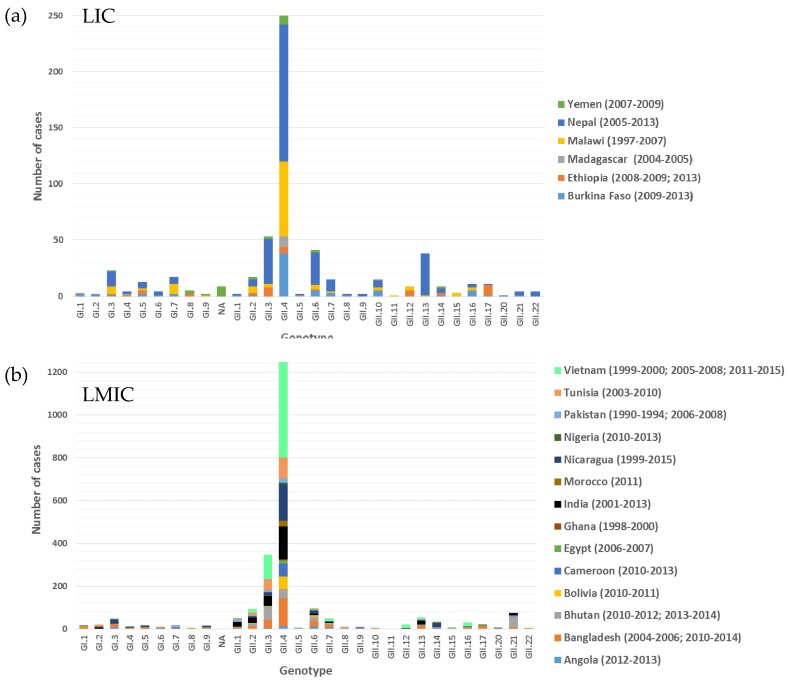
(**a**) The distribution of norovirus capsid genotypes detected in six LIC between 1997 and 2013; (**b**) the distribution of norovirus capsid genotypes circulating in 14 LMIC between 1990 to 1994 and 1998 to 2015. The years in which the studies were conducted are indicated in parenthesis after each country. NA = not assigned a GI genotype.

**Figure 5 viruses-11-00341-f005:**
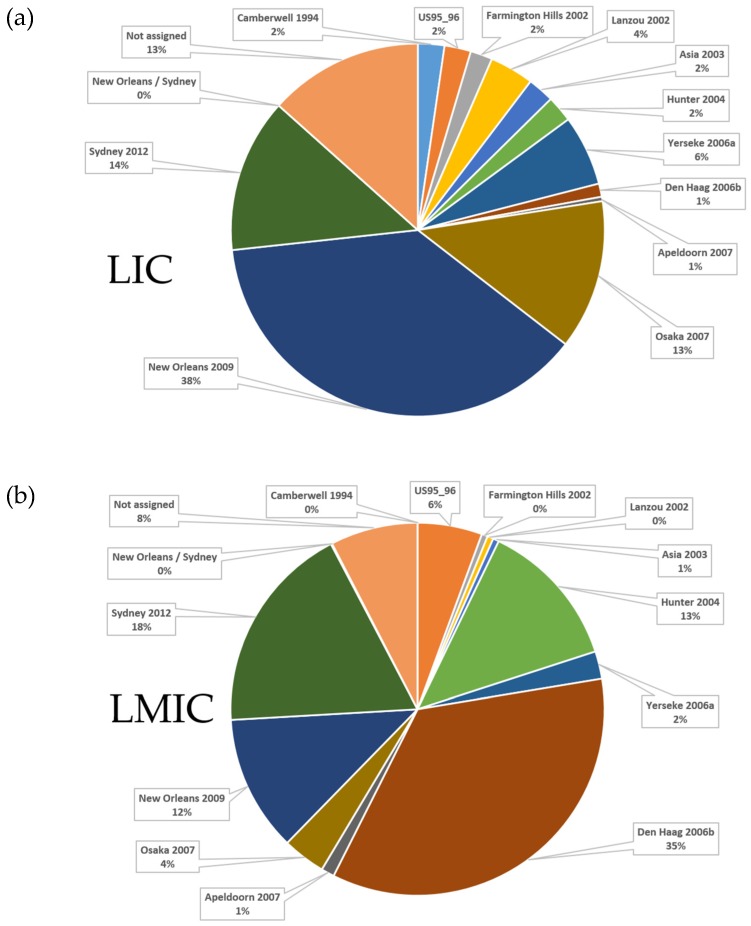
The distribution of norovirus GII.4 variants in (**a**) LIC (5 countries, 9 studies, and 167 typed variants) and (**b**) LMIC (14 countries, 25 studies, and 945 typed variants) between 1997 and 2015. The GII.4 variant was determined based on partial capsid genotyping.

**Table 1 viruses-11-00341-t001:** The detection of norovirus in LIC and LMIC in gastroenteritis cases and asymptomatic controls.

Group	No. of Studies/ No. of Countries	Number of Cases Tested	Number of Norovirus +	Norovirus Detection % (95% CI)
**LIC**				
<5 years inpatient	9/6	6713	733	11 (10–12)
<5 years outpatient	5/4	2430	226	9 (8–11)
Mixed age inpatient	1/1	229	95	42 (35–48)
Mixed age and setting	3/3	994	127	13 (11–15)
Overall symptomatic cases	15/8	10366	1181	11 (11–12)
Asymptomatic controls	7/6	1903	166	9 (8–10)
**LMIC**				
<5 years inpatient	30/13	13228	2294	17 (17–18)
<5 years outpatient	8/4	1058	123	12 (10–14)
<18 years inpatient	6/6	1603	142	9 (8–10)
Mixed age and setting	14/10	9969	1401	14 (13–15)
Overall symptomatic cases	49/21	25922	3977	15 (15–16)
Asymptomatic controls	16/11	3407	257	8 (8–10)

+ Number of norovirus positives.

**Table 2 viruses-11-00341-t002:** The prevalence of norovirus GI, GII, and GIV in LIC and LMIC in symptomatic norovirus cases and asymptomatic controls.

Setting	Norovirus Positives	GI n (%)	GII n (%)	GI + GII n (%)	GIV *
**LIC**					
Symptomatic	694	82 (11.8)	606 (87.3)	6 (0.9)	nt
Asymptomatic	139	30 (21.6)	108 (77.7)	1 (0.7)	nt
**LMIC**					
Symptomatic	3169	425 (13.4)	2642 (83.4)	86 (2.7)	16 (0.5)
Asymptomatic	136	33 (24)	99 (73)	4 (3)	0

* Three studies from Bangladesh detected norovirus GIV in 16/566 norovirus-positive specimens. nt = not tested.

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
