# Peer review of "Norovirus Infections and Disease in Lower-Middle- and Low-Income Countries, 1997–2018"

_viruses, 2019, doi:10.3390/v11040341_

Round 1
Reviewer 1 Report
The systematic review of norovirus infection in lower-middle and low income countries provided in the manuscript represents a welcome addition to the literature on the topic since there are far fewer studies from these regions and since there are large variations between studies, making it difficult to obtain a clear overview of the situation in those countries of great epidemiological importance. Overall, the review appears well conducted and its conclusions represent the best of what can be obtained at the moment given the intrinsic limitations of the available literature. By including recent publications, it largely confirms conclusions obtained from earlier systematic studies and brings to light some new information such as the high prevalence of co-infections in these countries, the large diversity of circulating strains despite the dominance of GII.4 and of GII.3 in the case of young children, or the uncertain seasonality. It also points out the importance of missing information regarding molecular-based data from many low resource countries.
Specific comments:
- In the materials and methods section, exclusion from the search strategy of the term outbreak in the title should be explained and justified.
- Line 159, the prevalence of GI in asymptomatic controls is 23%, whilst it is 21,6% in Table 2.
- Although only few studies report on symptoms severity, high prevalence of severe and mild dehydration were reported in several studies (lines 174-175). Is a comparison with rotavirus severity in the same settings possible? GE caused by norovirus is generaly considered milder than that caused by rotavirus. However, this is largely derived from studies in high income countries. A potential higher severity of norovirus GE in low resources countries may also be suggested by the significant higher prevalence of norovirus among inpatients in comparison with outpatients. A discussion on this matter would be a nice addition with regard to the importance of vaccine development.
Author Response
Reviewer 1
Specific comments:
Comment:
- In the materials and methods section, exclusion from the search strategy
of the term outbreak in the title should be explained and justified.
Response:
The sentence was modified on Page 2, lines 83-8:
The search filter was set to exclude hits containing the term “outbreak” or “outbreaks” in the title (Supplementary File 1), since the review was aimed at sporadic norovirus disease.
Comment:
- Line 159, the prevalence of GI in asymptomatic controls is 23%, whilst it is
21,6% in Table 2.
Response:
The mistake has been corrected. Page 5, line 161-162:
Increased detection of norovirus GI was reported in asymptomatic controls in both LIC and LMIC (21.6% and 24% vs 12% and 13% in symptomatic cases)
Comment:
- Although only few studies report on symptoms severity, high prevalence of
severe and mild dehydration were reported in several studies (lines 174-175).
Is a comparison with rotavirus severity in the same settings possible? GE
caused by norovirus is generaly considered milder than that caused by
rotavirus. However, this is largely derived from studies in high income
countries. A potential higher severity of norovirus GE in low resources
countries may also be suggested by the significant higher prevalence of
norovirus among inpatients in comparison with outpatients. A discussion on
this matter would be a nice addition with regard to the importance of vaccine
development.
Response:
Thank you for the suggestion. Four papers provided data on dehydration in single rotavirus and norovirus infections. The following section was included in the Discussion to address the comment:
Page 10, Lines 317-324
Four studies provided comparative data on dehydration for norovirus and rotavirus. In two of these, rotavirus caused dehydration in a higher percentage of patients (64% vs 18.9% [77] and 56% vs 30% [71]) than norovirus. In a study in Ethiopia, norovirus caused dehydration in 19% of single norovirus infections compared to only 5% of rotavirus infections that led to severe dehydration [48]. Another study in Zambia reported comparable levels of mild and moderate dehydration for patients infected with rotavirus or norovirus, but no severe dehydration [96]. Thus the limited available data suggest that norovirus could display increased severity in some settings but in general rotavirus causes more severe disease.
Additional changes:
Upon revisiting the papers for rotavirus dehydration information, three additional studies that provided dehydration data on single norovirus infections were identified and incorporated in the results section and discussion.
Page 6, lines 176-178
Six studies reported severe dehydration in the following percentages of single norovirus infections, 15.5% [26], 18.9% [77, 97], 19% [48], 19.4% [38] and 30.3% [71].
Page 10, lines 315-316
Six studies reported severe dehydration in 15-30% of single norovirus positive cases…….

Reviewer 2 Report
Comments for “Systematic review of noroviruses in lower-middle income and low income countries” by Janet Mans
The review article entitled “Systematic review of noroviruses in lower-middle income and low income countries” by Dr. Janet Mans is an extremely important addition to the norovirus literature that aims to clarify the prevalence and distribution of noroviruses in LIC and LMIC nations, filling a void in our knowledge of the epidemiology in these global regions. A review of this focus and scope is currently needed, since the most recent analyses are from 2008 and 2014. Overall, the review is very well organized and thorough with a very sound and clear methodology. In particular, the author analyzes issues like co-infection and seasonality, which are important aspects of norovirus disease that have not been well defined but that are important to consider. Additionally, the author discusses the topic of diagnosis of norovirus infection “with or without vomiting”, which is an overlooked and underappreciated problem in the clinical assessment of gastroenteritis, and norovirus in particular. Thus, the comments below are predominantly technical issues to improve clarity in how some of the data are reported, along with some minor grammatical/punctuation suggestions, and only one area of addition that may be considered.
Major comments:
The only issue that I would suggest the author add, and only briefly, would be several sentences to explain how the genotypes and the variants are distinguished. In other words, in Figure 4, the legend states that the genotypes are based on viral capsid. Even the norovirus expert might appreciate a short reminder/explanation of how the genotypes and strain variants are distinguished from each other. Only two to three sentences for each should be sufficient – a bit of clarification here would be interesting and helpful.
2. The previous item brings up the issue of diagnostics. In the Discussion, the author mentioned “improved detection via real time RT-PCR…versus conventional RT-PCR (lines 286-287). I would change the terminology to “quantitative versus qualitative” RT-PCR (i.e., RT-qPCR versus RT-PCR). The “RT” in the technique stands for “reverse transcriptase”, but “Real-Time” also, unfortunately, can be abbreviated as RT, and it causes no end to confusion to a general audience.
3. One major formatting issue that needs to be taken care of is removal of dashes (-) when separating percentages from 95% CI. For example, looking at lines 144 through 146: in line 144, the phrase, “…gastroenteritis symptoms and in 8% (95% CI 7-9) of asymptomatic….” appears. This is very clear and easy to read. But in lines 145-146, there are parentheticals that are formatted as follows: “…(15% - 95% CI 15-16, …). The dash after 15% can be read as a separator OR as implicating a range. It is natural for the eye to detect dashes as indicating a range, and it makes reading the parentheticals very awkward. The reader has to re-train themselves to clarify that 15% - 95% is not a range, but rather, it is 15% which has a 95% CI of x. I would either replace the dashes with a comma or a semicolon – but PLEASE remove the dashes before 95% CI, since CI is followed by another dashed item, which is an actual range. These are present throughout the manuscript, and it would greatly enhance clarity if they were modified.
4. In the legend to Figure 2, it is stated that the “…number of studies are indicated in parentheses.” This is inaccurate, because the Confidence Intervals in the figure itself are in parentheses, but the citations for the studies are in brackets. This is a very important distinction, because brackets are not parentheses, and it will improve accuracy and help the reader to use the correct term.
5. Please revisit the legend to Figure 1. I believe that there is a sentence that reads: “Additional studies from that tested rotavirus-negative…”. This is awkward – please fix.
6. Also regarding the Figure 1 legend, in the sentence “Studies in LMIC had a median population size…”. Should this be “sample size”? I’m not entirely sure, but I think the accurate term should be “sample”. Please investigate.
7. In the results section lines 112 – 115, can the author please add one sentence that rounds out the explanation of Figure 1. Please discuss the 48 studies with genotype analysis and 64 qualitative studies that are in the final boxes in Figure 1. This needs a bit of clarification in the text, because the way the Figure is outlined, it looks as if, out of the 106 studies that were assessed for eligibility, when 42 were removed, we are left with 64 + 48. The box with 106 branches off into separate boxes indicating 48 and 64 studies that were included, which adds up to 112. With a total of three arrows coming off of the box with n=106, it suggests that there are three unique subsets within that 106, but the math doesn’t add up. Please clarify. Also, regarding Figure 1, you might want to emphasize the word “Excluded” in the two boxes that have excluded studies, since the bottom box partially aligns with the “included” heading on the left hand side of the figure. Or put a different border around those boxes? I hope that made sense – please revisit.
8. Table 2 is very clear, but in the text, the author talks about GI/GII ratios, without ever actually indicating ratios (higher or lower for which genotypes?). Ratios can be confusing to think about, and so I am hesitant to suggest adding a column in the table to indicate the ratios. However, if ratios are mentioned by name, then an actual number or trend should be indicated somewhere, in the text and/or in the table. Alternatively, the term “ratio” could be removed, and just state that there is a difference in GII versus GI diagnoses in children that were the same in LIC and LMIC settings.
9. Question about Table 2. In the text, it is suggested that Table 2 is assessing children, but that is nowhere in the title of the table or on the table itself. Are these data for children only? Please clarify.
10. In the legend to Figure 4(b), should the number 17 actually be 14? I think that is a typo, since the figure shows 14 countries, and text in line 236 says “14 countries”.
11. Is it possible to add the labels LIC and LMIC to the graphs in Figure 4, as they are nicely prominent in Figure 5? It would make it a bit easier to read, and bring it in alignment with Figure 5.
Minor comments:
Below are minor comments simply pointing out some typographical errors or suggestions for grammatical clarification:
• Lines 18-19 of the abstract. Either separate into two sentences (i.e., a period after the word “overall”, or add the word “and” after the word “overall”).
• Line 37: The word “dominated” is awkward here. I would probably modify this sentence to “…have emerged and are predominant in certain regions…”, or “…have emerged and currently dominate…”
• Line 51: The sentence reads: “…prevalence estimates are based upon originated from high and…” Please remove either “based upon” or “originated from”, or add the word “and” in between them.
• Line 72: “…low LICs” is redundant. “low” is already implied in LIC.
• Line 107: Should read “Statistically significant”
• Lines 168 – 170: This should probably be broken up into two sentences, the first sentence ending after the word “fever”.
• Line 230: The comma does not need to be after the word “review”.
• Figures 4 and 5. Is it possible to move the (a) and (b) labels to the upper left region of the images? This is very minor – if it is too difficult in the software, forget about it. But it is sort of unconventional to put the letter labels below the images.
• Line 289: “The percentage norovirus…” Please add the word “of” before norovirus.
• Lines 306-307: Please either break this into two sentences, or add the word “and” before the word “similar”.

Author Response
Reviewer 2:
Major comments:
Comment
The only issue that I would suggest the author add, and only briefly, would be several sentences to explain how the genotypes and the variants are distinguished. In other words, in Figure 4, the legend states that the genotypes are based on viral capsid. Even the norovirus expert might appreciate a short reminder/explanation of how the genotypes and strain variants are distinguished from each other. Only two to three sentences for each should be sufficient – a bit of clarification here would be interesting and helpful.
Response:
Thank you for the suggestion. The following sentences were added in the section on genotyping:
Page 7, lines 219-220
Norovirus genotypes are classified with a phylogeny-based dual typing system using partial polymerase and complete capsid nucleotide sequences (i.e. GII.Pe/GII.4) [100].
Page 8, lines 247-249
Norovirus GII.4 variants are distinguished based on phylogenetic analysis of the complete capsid gene [100]. New variants are recognised once they have been established in at least two separate geographic locations [100].
Comment:
2. The previous item brings up the issue of diagnostics. In the Discussion, the author mentioned “improved detection via real time RT-PCR…versus conventional RT-PCR (lines 286-287). I would change the terminology to “quantitative versus qualitative” RT-PCR (i.e., RT-qPCR versus RT-PCR). The “RT” in the technique stands for “reverse transcriptase”, but “Real-Time” also, unfortunately, can be abbreviated as RT, and it causes no end to confusion to a general audience.
Response:
The terminology was changed as suggested. Page 10, lines 277-278.
This may also reflect improved detection via quantitative RT-PCR as four of the five studies used quantitative versus qualitative RT-PCR [20, 22, 27, 81] which was used in many earlier studies in LIC.
Comment:
3. One major formatting issue that needs to be taken care of is removal of dashes (-) when separating percentages from 95% CI. For example, looking at lines 144 through 146: in line 144, the phrase, “…gastroenteritis symptoms and in 8% (95% CI 7-9) of asymptomatic….” appears. This is very clear and easy to read. But in lines 145-146, there are parentheticals that are formatted as follows: “…(15% - 95% CI 15-16, …). The dash after 15% can be read as a separator OR as implicating a range. It is natural for the eye to detect dashes as indicating a range, and it makes reading the parentheticals very awkward. The reader has to re-train themselves to clarify that 15% - 95% is not a range, but rather, it is 15% which has a 95% CI of x. I would either replace the dashes with a comma or a semicolon – but PLEASE remove the dashes before 95% CI, since CI is followed by another dashed item, which is an actual range. These are present throughout the manuscript, and it would greatly enhance clarity if they were modified.
Response:
Thank you for the suggestion, it definitely enhances clarity. The dashes were removed before the 95% CI and replaced by a semicolon throughout the text, these are all indicated by track changes throughout the revised manuscript.
Comment:
4. In the legend to Figure 2, it is stated that the “…number of studies are indicated in parentheses.” This is inaccurate, because the Confidence Intervals in the figure itself are in parentheses, but the citations for the studies are in brackets. This is a very important distinction, because brackets are not parentheses, and it will improve accuracy and help the reader to use the correct term.
Response:
Thank you for pointing out the mistake that would have caused a lot of confusion. The word parentheses was changed to brackets in the legend to Figure 2 on Page 4, line 135.
Comment:
5. Please revisit the legend to Figure 1. I believe that there is a sentence that reads: “Additional studies from that tested rotavirus-negative…”. This is awkward – please fix.
Response:
This sentence was in the paragraph before Figure 2 (original lines 121-123) and the word “countries”was missing. It has been revised to the following:
Page 4, line 124-126
“Additional studies, from countries which tested pathogen-negative specimens for noroviruses, were included in the world map with their respective norovirus prevalence”.
Comment:
6. Also regarding the Figure 1 legend, in the sentence “Studies in LMIC had a median population size…”. Should this be “sample size”? I’m not entirely sure, but I think the accurate term should be “sample”. Please investigate.
Response:
This sentence was also in the paragraph preceding Figure 2. The word population was replaced with sample.
Page 4, line 128. “Studies in LMIC had a median sample size of 265 and ranged from 34 to 2495 participants”.
Comment:
7. In the results section lines 112 – 115, can the author please add one sentence that rounds out the explanation of Figure 1. Please discuss the 48 studies with genotype analysis and 64 qualitative studies that are in the final boxes in Figure 1. This needs a bit of clarification in the text, because the way the Figure is outlined, it looks as if, out of the 106 studies that were assessed for eligibility, when 42 were removed, we are left with 64 + 48. The box with 106 branches off into separate boxes indicating 48 and 64 studies that were included, which adds up to 112. With a total of three arrows coming off of the box with n=106, it suggests that there are three unique subsets within that 106, but the math doesn’t add up. Please clarify. Also, regarding Figure 1, you might want to emphasize the word “Excluded” in the two boxes that have excluded studies, since the bottom box partially aligns with the “included” heading on the left hand side of the figure. Or put a different border around those boxes? I hope that made sense – please revisit.
Response:
The following sentences were added to clarify Figure 1:
Page 3, lines 116-119
“Sixty four studies were included in the review of norovirus prevalence. Forty three studies in total were included for norovirus genotype analysis. Of these, 37 were studies that reported both prevalence and genotype data and 6 were studies that reported genotyping data from noroviruses detected in pathogen-negative specimens.”
Page 3, line 120: Figure 1 was revised to correctly reflect the screening/selection of studies.
Thank you for pointing out that the figure needs revision. I realise that the way it was presented is misleading and the figure was revised to clearly show that 64/106 papers were included in the qualitative synthesis; 42/106 papers were excluded with reasons and that 43 papers were included in the genotype analysis (37 from the 64 that was included in the prevalence analysis and 6 studies that screened pathogen-negative specimens and presented genotype data). The word “Excluded” was capitalised and changed to bold to emphasize that those boxes contain the excluded studies.
Comment:
8. Table 2 is very clear, but in the text, the author talks about GI/GII ratios, without ever actually indicating ratios (higher or lower for which genotypes?). Ratios can be confusing to think about, and so I am hesitant to suggest adding a column in the table to indicate the ratios. However, if ratios are mentioned by name, then an actual number or trend should be indicated somewhere, in the text and/or in the table. Alternatively, the term “ratio” could be removed, and just state that there is a difference in GII versus GI diagnoses in children that were the same in LIC and LMIC settings.
Response:
The word ratio was removed and the sentence was changed as follows on Page 5, lines 160-161:
The prevalence of norovirus GI in relation to GII was similar in symptomatic children in both settings (Table 2).
In the Discussion the sentence on page 10, line 300-301 was also revised to remove the word “ratio”:
Similar GI and GII distributions were reported (GI/GII/mixed - 12/87/0.9, 13/83/3) in symptomatic infections.
Comment:
9. Question about Table 2. In the text, it is suggested that Table 2 is assessing children, but that is nowhere in the title of the table or on the table itself. Are these data for children only? Please clarify.
Response:
The data are from all the studies that reported GI and GII numbers, thus including mixed ages. Therefor the reference to children in the text is wrong and was revised.
Page 5, lines 160-162:
“The prevalence of norovirus GI in relation to GII was similar in symptomatic infections in both settings (Table 2). Increased detection of norovirus GI was reported in asymptomatic controls in both LIC and LMIC (21.6% and 24% vs 12% and 13% in symptomatic cases)”.
Comment:
10. In the legend to Figure 4(b), should the number 17 actually be 14? I think that is a typo, since the figure shows 14 countries, and text in line 236 says “14 countries”.
Response:
The mistake was corrected, it should be 14 countries. Page 8, line 244.
Comment:
11. Is it possible to add the labels LIC and LMIC to the graphs in Figure 4, as they are nicely prominent in Figure 5? It would make it a bit easier to read, and bring it in alignment with Figure 5.
Response:
The requested labels were added to Figure 4 on Page 8.
Minor comments:
Below are minor comments simply pointing out some typographical errors or suggestions for grammatical clarification:
• Lines 18-19 of the abstract. Either separate into two sentences (i.e., a period after the word “overall”, or add the word “and” after the word “overall”).
Response: The sentences were separated as suggested by the reviewer. Page 1, line 19.
• Line 37: The word “dominated” is awkward here. I would probably modify this sentence to “…have emerged and are predominant in certain regions…”, or “…have emerged and currently dominate…”
Response: The sentence was revised as suggested. Page 1, line 35.
“Recently, other genotypes such as GII.17 and GII.2 have emerged and are predominant in certain regions of the world.”
• Line 51: The sentence reads: “…prevalence estimates are based upon originated from high and…” Please remove either “based upon” or “originated from”, or add the word “and” in between them.
Response: The sentence was revised as follows: Page 2, lines 50-52.
“The majority of data used to estimate global norovirus prevalence emanated from studies performed in high and upper-middle income countries.”
• Line 72: “…low LICs” is redundant. “low” is already implied in LIC.
Response: The mistake was corrected. Page 2, line 72.
• Line 107: Should read “Statistically significant”
Response: Corrected as suggested. Page 3, line 108.
• Lines 168 – 170: This should probably be broken up into two sentences, the first sentence ending after the word “fever”.
Response: Corrected as suggested. Page 6, line 172.
• Line 230: The comma does not need to be after the word “review”.
Response: The comma was removed. Page 7, line 224.
• Figures 4 and 5. Is it possible to move the (a) and (b) labels to the upper left region of the images? This is very minor – if it is too difficult in the software, forget about it. But it is sort of unconventional to put the letter labels below the images.
Response: The (a) and (b) labels in Figures 4 and 5 were moved to the upper left region of the images as suggested. Page 8-9.
• Line 289: “The percentage norovirus…” Please add the word “of” before norovirus.
Response: Changed as suggested. Page 10, line 280
• Lines 306-307: Please either break this into two sentences, or add the word “and” before the word “similar”.
Response: The sentence was split in two as suggested. Page 10, lines 300-301.
Norovirus GII was the predominant genogroup detected in all studies in LIC and LMIC. Similar GI and GII distributions were reported (GI/GII/mixed - 12/87/0.9, 13/83/3) in symptomatic infections.
Reviewer 3 Report
Systematic review of noroviruses in lower-middle income and low income countries
By Janet Mans
Submitted to Viruses (Editorial No. viruses-479095)
General Comments
This review on the significance of norovirus (NoV) infections and disease with particular emphasis on low- and middle-low income countries is timely and topical. The collection of relevant publications has been comprehensive, and the evaluation is well organized. The English is very good. Some additional references have been suggested, which are considered to provide additional information to the epidemiological framework.
Specific Comments
Line
2 Reconsider Title, e.g., ‘Norovirus infections and disease in lower-middle and low-income countries, 1997-2018’, or similar.
9 Consider reading: … Noroviruses are a major cause… [taking into account that there is a large variety of different NoV genotypes co-circulating. – Consider rephrasing throughout manuscript]
12 … and LMICs (World Bank classification).
15 … of the gastroenteritis patients and…
19 … predominant genotype in all settings…
31 Noroviruses form a genus of the Caliciviridae…
34 … sub-divided into …
38 … global burden of norovirus illness…
39 … vaccine design. [Since this item comes up in Conclusions again, it is suggested to provide a reference, e.g. Cortes-Penfield NW, Ramani S, Estes MK, Atmar RL. Prospects and Challenges in the Development of a Norovirus Vaccine. Clin Ther. 2017 Aug;39(8):1537-1549.]
43 Consider reading: … Norovirus-associated infections are estimated… cases of disease and … deaths…
51 … The majority of published data…
53 … noroviruses as enteropathogens…
65 … children of<11 months of age… [Consider analogous changes in other places of the ms.]
71 … noroviruses are important pathogens globally…
74 … 64 studies of which 17 (…) were published after March 2016 and have not been…
223 … according to Tra My and colleagues… [Please confirm.]
241 Supplementary Table 1. This can be omitted, since the data are mostly covered by the Text.
279f Consider citation some papers on the age-dependent sero-epidemiology of NoV infections which strongly suggest that the rate of asymptomatic infections in infancy and early childhood is high.
339 … This would be expected for pathogens that independently cause diarrhea… This statement is contradicted by much of the data and should be rephrased.
359 High frequency of overall detection of NoV GII.4 genotype. This fact is known since some time and has been analysed in more detail by the group of Lindesmith/Mallory/ Debbink/Baric and colleagues. Consider to cite some of their publications, e.g.,
Mallory ML, Lindesmith LC, Graham RL, Baric RS. GII.4 Human Norovirus: Surveying the Antigenic Landscape. Viruses. 2019 Feb 20;11(2). pii: E177.
Lindesmith LC, Brewer-Jensen PD, Mallory ML, Yount B, Collins MH, Debbink K, Graham RL, Baric RS. Human Norovirus Epitope D Plasticity Allows Escape from Antibody Immunity without Loss of Capacity for Binding Cellular Ligands. J Virol. 2019 Jan 4;93(2). pii: e01813-18.
Lindesmith LC, Mallory ML, Debbink K, Donaldson EF, Brewer-Jensen PD, Swann EW, Sheahan TP, Graham RL, Beltramello M, Corti D, Lanzavecchia A, Baric RS. Conformational Occlusion of Blockade Antibody Epitopes, a Novel Mechanism of GII.4 Human Norovirus Immune Evasion. mSphere. 2018 Feb 7;3(1). pii: e00518-17.
366 Recent emergence of GII.17 NoV strains. Consider citation of 2-3 publications, e.g.,
Lindesmith LC, Kocher JF, Donaldson EF, Debbink K, Mallory ML, Swann EW, Brewer-Jensen PD, Baric RS. Emergence of Novel Human Norovirus GII.17 Strains Correlates With Changes in Blockade Antibody Epitopes. J Infect Dis. 2017 Dec 5;216(10):1227-1234.
Qian Y, Song M, Jiang X, Xia M, Meller J, Tan M, Chen Y, Li X, Rao Z. Structural Adaptations of Norovirus GII.17/13/21 Lineage through Two Distinct Evolutionary Paths. J Virol. 2018 Dec 10;93(1). pii: e01655-18.
Sang S, Yang X. Evolutionary dynamics of GII.17 norovirus. PeerJ. 2018 Feb 1;6:e4333.
389 … that noroviruses are important pathogens in low-income settings. ..
Author Response
Reviewer 3:
Specific Comments
Line
2 Reconsider Title, e.g., ‘Norovirus infections and disease in lower-middle and low-income countries, 1997-2018’, or similar.
Response:
The title was changed to:
“Norovirus infections and disease in lower-middle and low-income countries, 1997- 2018”
9 Consider reading: … Noroviruses are a major cause… [taking into account that there is a large variety of different NoV genotypes co-circulating. – Consider rephrasing throughout manuscript]
Response:
The suggested changes, indicated by track changes, were made on Page 1 (lines 9 and 27), Page 2 (lines 43 and 71) and Page 12, line 394.
12 … and LMICs (World Bank classification).
Response:
Added to the abstract.
15 … of the gastroenteritis patients and…
Response:
Revised in abstract.
19 … predominant genotype in all settings…
Response:
Revised in abstract.
31 Noroviruses form a genus of the Caliciviridae…
Response:
Revised on Page 1, lines 29-30
34 … sub-divided into …
Response:
Revised as suggested, Page 1, lines 32
38 … global burden of norovirus illness…
Response:
Revised as suggested, Page 1, line 36.
39 … vaccine design. [Since this item comes up in Conclusions again, it is suggested to provide a reference, e.g. Cortes-Penfield NW, Ramani S, Estes MK, Atmar RL. Prospects and Challenges in the Development of a Norovirus Vaccine. Clin Ther. 2017 Aug;39(8):1537-1549.]
Response:
The suggested reference was added. Page 1, line 12.
43 Consider reading: … Norovirus-associated infections are estimated… cases of disease and … deaths…
Response:
The sentence was changed as suggested: Page 1, lines 42-43
Norovirus-associated infections are estimated to annually cause 677 million cases of disease and approximately 200 000 deaths globally
51 … The majority of published data…
Response:
The sentence was changed as follows on Page 2, lines 50-51:
The majority of data used to estimate global norovirus prevalence emanated from studies performed in high and upper-middle income countries.
53 … noroviruses as enteropathogens…
Response:
Changed as suggested on Page 2, lines 52-54:
In recent years there has been an increase in studies from low resource settings [20-28], reflecting growing awareness of the importance of noroviruses as enteropathogens.
65 … children of<11 months of age… [Consider analogous changes in other places of the ms.]
Response:
The suggested changes were made throughout the manuscript as indicated by track changes.
71 … noroviruses are important pathogens globally…
Response:
Suggested changes made on Page 2, line 71.
74 … 64 studies of which 17 (…) were published after March 2016 and have not been…
Response:
Changes made on Page 2, lines 74-75
223 … according to Tra My and colleagues… [Please confirm.]
Response:
Page 7, line 215: The author name was changed from Phan to Tra My.
241 Supplementary Table 1. This can be omitted, since the data are mostly covered by the Text.
Response:
I omitted the reference to Supplementary File 1 in lines 229 and 234, but moved it to line 221 to provide the reader with information on the studies that were analysed in terms of genotypes.
279f Consider citation some papers on the age-dependent sero-epidemiology of NoV infections which strongly suggest that the rate of asymptomatic infections in infancy and early childhood is high.
Response:
Thank you for the suggestion. The following sentences and reference were added on Page 10, lines 284-287:
These data are in agreement with a sero-epidemiological study in Uganda [104], which detected norovirus antibodies in 77% of children by 1 year of age and in 99% of two-year old children and strongly suggest that the rate of asymptomatic infections in infancy and early childhood is high in low resource settings.
339 … This would be expected for pathogens that independently cause diarrhea… This statement is contradicted by much of the data and should be rephrased.
Response:
The statement referred to the results of Andersson et al and was rephrased to clarify that statistically it is expected that two pathogens that cause diarrhoea independently would be negatively associated in patients with diarrhoea, but not in control patients.
Page 11, lines 338-341:
Andersson and co-workers [109] found a negative association (co-infections less common than expected from probability) between rotavirus and norovirus GII in symptomatic patients. Statistically, pathogens that independently cause diarrhoea are expected to be negatively associated in symptomatic patients [109].
359 High frequency of overall detection of NoV GII.4 genotype. This fact is known since some time and has been analysed in more detail by the group of Lindesmith/Mallory/ Debbink/Baric and colleagues. Consider to cite some of their publications, e.g.,
Mallory ML, Lindesmith LC, Graham RL, Baric RS. GII.4 Human Norovirus: Surveying the Antigenic Landscape. Viruses. 2019 Feb 20;11(2). pii: E177.
Lindesmith LC, Brewer-Jensen PD, Mallory ML, Yount B, Collins MH, Debbink K, Graham RL, Baric RS. Human Norovirus Epitope D Plasticity Allows Escape from Antibody Immunity without Loss of Capacity for Binding Cellular Ligands. J Virol. 2019 Jan 4;93(2). pii: e01813-18.
Lindesmith LC, Mallory ML, Debbink K, Donaldson EF, Brewer-Jensen PD, Swann EW, Sheahan TP, Graham RL, Beltramello M, Corti D, Lanzavecchia A, Baric RS. Conformational Occlusion of Blockade Antibody Epitopes, a Novel Mechanism of GII.4 Human Norovirus Immune Evasion. mSphere. 2018 Feb 7;3(1). pii: e00518-17.
Response:
The following sentences were added to the Discussion on Page 11, lines 372-375:
The well-known predominance of GII.4 may be related to its ability to evade the immune system by conformational occlusion of blockade antibody epitopes [114] and variation in epitopes that allow escape from immunity without impairing binding to cellular ligands [115].
The two Lindesmith papers below were cited:
Lindesmith LC, Brewer-Jensen PD, Mallory ML, Yount B, Collins MH, Debbink K, Graham RL, Baric RS. Human Norovirus Epitope D Plasticity Allows Escape from Antibody Immunity without Loss of Capacity for Binding Cellular Ligands. J Virol. 2019 Jan 4;93(2). pii: e01813-18.
Lindesmith LC, Mallory ML, Debbink K, Donaldson EF, Brewer-Jensen PD, Swann EW, Sheahan TP, Graham RL, Beltramello M, Corti D, Lanzavecchia A, Baric RS. Conformational Occlusion of Blockade Antibody Epitopes, a Novel Mechanism of GII.4 Human Norovirus Immune Evasion. mSphere. 2018 Feb 7;3(1). pii: e00518-17.
366 Recent emergence of GII.17 NoV strains. Consider citation of 2-3 publications, e.g.,
Lindesmith LC, Kocher JF, Donaldson EF, Debbink K, Mallory ML, Swann EW, Brewer-Jensen PD, Baric RS. Emergence of Novel Human Norovirus GII.17 Strains Correlates With Changes in Blockade Antibody Epitopes. J Infect Dis. 2017 Dec 5;216(10):1227-1234.
Qian Y, Song M, Jiang X, Xia M, Meller J, Tan M, Chen Y, Li X, Rao Z. Structural Adaptations of Norovirus GII.17/13/21 Lineage through Two Distinct Evolutionary Paths. J Virol. 2018 Dec 10;93(1). pii: e01655-18.
Sang S, Yang X. Evolutionary dynamics of GII.17 norovirus. PeerJ. 2018 Feb 1;6:e4333.
Response:
Citations were added for GII.17 and for GII.2, Page 11, lines 367-369.
Since the studies that form part of this review were performed between 1997 and 2015, the more recent emergence of novel GII.17 [11, 113] and GII.2 [10] in LIC and LMIC could not be evaluated.
389 … that noroviruses are important pathogens in low-income settings. ..
Response:
The suggested change was made on Page 12, line 394.